# It Depends: Understanding Why Models Struggle with Long-Range Dependencies

## Abstract

As researchers seek to better understand the architectures that have driven recent performance improvements in Natural Language Processing (NLP), analyses are commonly carried out on the handling of long-range dependencies. However, the use of this phrase can be inconsistent and unclear across the literature, making it difficult to understand which element of a dependency is the key factor affecting performance. In this work we create a systematic framework for discussing dependencies and carrying out careful analyses of how the many variables involved can impact architecture performance. By disentangling these factors we clarify the discussion of dependencies and enable more meaningful comparisons across architectures. Using this framework, our experiments find that, despite often being the main focus of work on dependencies, the distance between a token and the tokens on which it depends is not a substantial factor in model performance. However, the number of tokens involved in the dependency and the complexity and nature of the dependency are important factors. We also find that architectural elements do not uniformly improve or degrade performance across tasks, but that their effect is dependent on the nature of the dependency being modelled. This framework can be built on and used to motivate principled discussions of architecture performance in the future.

Long-range dependencies have been the subject of considerable attention recently in considering the performance characteristics of new and existing machine learning architectures (Tay et al., 2021; Orvieto et al., 2023). However, it is often unclear exactly what is intended when a researcher uses the term "long-range dependency." Indeed, for this reason, the exact nature of the reported difficulties associated with modeling long-range dependencies remains underexplored. For example, does difficulty arise purely from the *distance* between dependent elements in a string, or, rather, is it also a function of the *complexity* of the dependency that exists? Do different architectures struggle with different aspects of long-range dependencies? Understanding model performance in terms of these factors is crucial in order to make informed architectural choices and continue improving models' abilities to perform language-based tasks.

In this work, we craft a systematic framework for the evaluation of models on tasks containing long-range dependencies. This framework allows us to make detailed comparisons across models, as well as performing precise analysis of the effects of variables such as the distance covered by a dependency in order to answer some of the crucial outstanding questions on this topic. This is done with the aim of providing a clearer understanding of how these factors influence the performance of various modelling architectures when handling dependencies. Our framework consists of framing dependencies as a formal language completion task, where we can vary the number of tokens that the completion depends on, how distant those tokens are in the sequence, and the relationship that exists between the completion and the tokens that it depends on. We create datasets of this form and use them to evaluate a variety of architectures.

We find that, despite being among the most prominent factors in discussions of long-range dependencies, the distance spanned by the dependency actually has little impact on task performance across all architectures tested. The most significant factor in the difficulty of predicting a token is, in fact, the number of tokens upon which the token depends, with performance across all architectures and all tasks deteriorating significantly as the length of the dependency sequence increases. This effect on performance was not alleviated by increasing model size. The complexity of the dependency involved also had a significant impact, with a non-uniform effect across architectures. Characterizing

the complexity of these dependencies is difficult, and we do not find that any of the several existing characterizations that we explored correspond neatly with our results.

We hope that this work will motivate a further discussion of the modelling of dependencies and will lead to further principled analysis on the properties of architectures currently used for language-based tasks.

# 1 WHAT IS A LONG-RANGE DEPENDENCY?

To discuss how models handle dependencies, we must first clearly define them. Say we are given some alphabet $\Sigma$ and a distribution over strings that defines a language, $p : \Sigma^* \to [0, 1]$, which is parameterised autoregressively, i.e. for $\mathbf{x} = x_1 \ldots x_T \in \Sigma^*$, $p(\mathbf{x}) = \prod_{t=1}^{T} p(x_t \mid \mathbf{x}_{<t})$. For a token $x_t \in \mathbf{x}$, we define its dependency distance as the minimal $k \in \mathbb{N}$ such that $p(x_t \mid \mathbf{x}_{<t}) = p(x_t \mid \mathbf{x}_{t-k+1:t})$. We define the dependency distance of a string $\mathbf{x} = x_1 \ldots x_T$ to be the maximum dependency distance of its tokens. We say that the language defined by the distribution contains long-range dependencies if there is no $K \in \mathbb{N}$ such that all $\mathbf{x} \in \Sigma^*$ have dependency distance smaller than $K$, i.e. there is no $K$ such that for any $\mathbf{x} \in \Sigma^*$ and any $t \in \{1, \ldots, |\mathbf{x}|\}$, $p(x_t \mid \mathbf{x}_{<t}) = p(x_t \mid \mathbf{x}_{t-K+1:t})$.

# 2 INVESTIGATING DEPENDENCIES

We set out to investigate how well models learn to model dependencies and how this is affected by a variety of factors. We do this by crafting a task based on artificial languages containing long-range dependencies. In our framework a language $\mathcal{L}(\Sigma, \alpha, \beta, f)$ is defined by an alphabet $\Sigma$, distinguished symbols $\alpha, \beta \in \Sigma$ and a function $f : \Sigma^* \to \Sigma^*$. A string $\mathbf{x} = x_1 \ldots x_{|\mathbf{x}|} \in \mathcal{L}(\Sigma, \Delta, \alpha, \beta, f)$ contains the distinguished symbol $\alpha$ at exactly two positions, $a_1$ and $a_2$, and the distinguished symbol $\beta$ at precisely two positions $b_1$ and $b_2 = |\mathbf{x}|$, with $a_1 < a_2 < b_1 < b_2$. Such a string satisfies $\mathbf{x}_{b_1+1:b_2-1} = f(\mathbf{x}_{a_1+1:a_2-1}) \in \Sigma^*$. We refer to $\mathbf{x}_{<a_1}$ as the *initial padding*, $\mathbf{p}$, $\mathbf{x}_{a_2+1:b_1-1}$ as the dependency padding, $\mathbf{d}$, and $\mathbf{x}_{a_1+1:a_2-1}$ as the *target sequence*, $\tilde{\mathbf{x}}$. In this work, a model is trained on complete strings belonging to $\mathcal{L}(\Sigma, \alpha, \beta, f)$ with a language modelling objective, and is evaluated by generating the completion of $\mathbf{x}_{<b_1+1}$. Since $\mathbf{x}_{b_1+1:b_2-1} = f(\tilde{\mathbf{x}})$, the desired completion will be dependent only on $\tilde{\mathbf{x}}$. For example, if we have $\mathcal{L}(\Sigma, \alpha, \beta, f)$ where $\Sigma = \{\alpha, \beta, a, b, c\}$ and $f$ is the identity function, $f(\mathbf{x}) = \mathbf{x}$, then an example of a string $\mathbf{x} \in \mathcal{L}(\Sigma, \alpha, \beta, f)$ is $\mathbf{x} = bcaaab\alpha abc\alpha bba\beta abc\beta$. A model being evaluated on this language may asked to generate a completion for $\mathbf{x} = bcaaab\alpha abc\alpha bba\beta$ (with the correct answer $abc\beta$).

## 2.1 RELEVANT FACTORS

This framework allows us to disentangle and measure the effect of multiple aspects of long-range dependencies:

**Dependency Gap.** We refer to the distance $D = b_1 - a_2 - 1$ as the *dependency gap*. By varying the dependency gap, $D$, we can investigate whether completion accuracy degrades when there is a dependency on information separated by many tokens.

**Target Sequence Length.** We refer to $T = |\tilde{\mathbf{x}}| = a_2 - a_1 - 1$ as the *target sequence length*. A string with a longer target sequence length results in a completion that is dependent on more elements, increasing the amount of information that the model must use and process. The *dependency distance* of the string, as defined in section 1, is given by $T + D + 2$. By separating this distance into two components we can precisely investigate whether the difficulty is due to the tokens involved in the dependency being very far away or whether it is due to there being a large number of them.

**Dependency Complexity.** The function $f$ determines the nature of the dependency on the target sequence $\tilde{\mathbf{x}}$. In the case of the identity function, this dependency is simple, but it may take the form of any function $f : \Sigma^* \to \Sigma^*$. By varying this function, we can understand when performance is affected not by the distance of a dependency, but by the complexity of the computation required. This

aspect was a primary focus on work by Delétang et al. (2023), but this work incorporates it as only one part of an experimental framework.

In practice, our experiments use distinguished symbols $\alpha = 1$, $\beta = 2$ and an alphabet $\Sigma$ containing these digits as well as all upper and lower case characters $a-z$ and $A-Z$. Below we give an example of a string used in our experiments. When testing, the part of the string highlighted in blue is given as input to the model and the part highlighted in red is the desired completion.

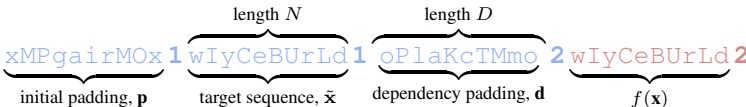

## 2.2 Choices for $f$

We use a variety of functions $f$ in order to investigate the effect of the nature of the dependency. These functions are described below and examples are given in table 2.

**Base.** For this task the desired completion is given by reproducing the target sequence, so $f = f_{\text{base}}$ where $f_{\text{base}}(x_1 \ldots x_n) = x_1 \ldots x_n$.

**Reverse.** Here $f = f_{\text{reverse}}$ where $f_{\text{reverse}}(x_1 \ldots x_n) = x_n \ldots x_1$, i.e. the string is reproduced in reverse.

**Even.** This task is dependent on the properties of the target sequence. If the length of the target sequence is even, it should be reproduced, while if it is odd it should be reproduced in reverse, i.e. $f = f_{\text{even}}$ where

$$f_{\text{even}}(x_1 \ldots x_n) = \begin{cases} f_{\text{base}}(x_1 \ldots x_n) = x_1 \ldots x_n & \text{if } n \in 2\mathbb{Z} \\ f_{\text{reverse}}(x_1 \ldots x_n) = x_n \ldots x_1 & \text{if } n \notin 2\mathbb{Z} \end{cases}$$

**Odd.** Define $f_{\text{count}} : \Sigma \times \Sigma^* \to \mathbb{N}_0$ so that $f_{\text{count}}(a, \mathbf{x}) = \sum_{i=0}^{n} \mathbb{1}\left[x_i = a\right]$, i.e. it returns the number of times that $a$ appears in $\mathbf{x}$. Additionally, define $f_{\text{odd count}} : \Sigma \times \Sigma^* \to \Sigma$

$$f_{\text{odd count}}(a, \mathbf{x}) = \begin{cases} a & \text{if } f_{\text{count}}(a, \mathbf{x}) \notin 2\mathbb{Z} \\ \text{empty string} & f_{\text{count}}(a, \mathbf{x}) \in 2\mathbb{Z} \end{cases}$$

Given $\mathbf{x} = x_1 \ldots x_n$, define $\mathcal{A}_{\mathbf{x}} = \{a_1, a_2, \ldots, a_m\}$ to be the set of characters appearing in $\mathbf{x}$, in the order they first appear, with repetitions removed, so $m \leq n$. Then, for this task, $f = f_{\text{odd}}$ where $f_{\text{odd}}(\mathbf{x}) = f_{\text{odd count}}(a_1) \ldots f_{\text{odd count}}(a_m)$. That is to say, each character appearing in $\mathbf{x}$ is reproduced, in order of its first appearance, if it appears an odd number of times in $\mathbf{x}$.

**Sort.** For this task $f = f_{\text{sort}}$ is the function that returns all of the characters of the input sequence sorted into alphabetical order.

**Palindrome.** For this task the input string $\mathbf{x}$ is a string obtained by initially creating a palindrome $\tilde{\mathbf{x}}$ such that $\tilde{\mathbf{x}} = \tilde{x}_1 \ldots \tilde{x}_{\frac{n}{2}} \tilde{x}_{\frac{n}{2}} \ldots \tilde{x}_1$ and subsequently removing one character. Then $f = f_{\text{palindrome}}$ is a function that returns the corrected palindrome.

**Dyck.** Inputs for this task are obtained by first obtaining a string belonging to a Dyck language over the set of alphabetical characters and subsequently removing one character. The function $f = f_{\text{dyck}}$ is then a function that returns a corrected Dyck string.

**Case.** This task requires switching the case of each alphabetical character. Define $c : \Sigma \to \Sigma$ such that

$$c(x) = \begin{cases} A & \text{if } x = a \\ \vdots & \vdots \\ z & \text{if } x = Z \end{cases}$$

i.e. $c$ swaps upper and lower case characters. Then $f = f_{\text{case}}$ where $f_{\text{case}}(x_1 \ldots x_n) = c(x_1) \ldots c(x_n)$.

**Periodic.** A target sequence in this language will take the form of a repeated subsequence, so $\mathbf{x} = x_1 \ldots x_n = \underbrace{(x_1 \ldots x_m) \ldots (x_1 \ldots x_m)}_{q \text{ repetitions}}$ where $n = qm$. Then $f = f_{\text{periodic}}$ where $f_{\text{periodic}}((x_1 \ldots x_m) \ldots (x_1 \ldots x_m)) = x_1 \ldots x_m$, i.e. it should reproduce the repeating sequence only once.

## 3 EXPERIMENTS

We conducted experiments using a number of architectures, all trained from scratch. This included transformers with and without positional encodings, LSTMs with and without attention, and Mamba state space models (Gu & Dao, 2024). Best hyperparameters for each architecture were found using a Bayesian optimisation procedure and are given in appendix C. Experiments were conducted separately for each of the 9 tasks described in section 2.2. For each task a dataset was generated containing 150,000 training examples, 10,000 validation examples and 10,000 test examples. In training and validation datasets, the dependency gap was no more than 16 and target sequence length was no more than 24. In testing the dependency gap could be up to 32, as could target sequence length. This allowed for analysis of not only performance across these lengths but additionally an examination of how performance changed for data that was out-of-distribution.

## 4 RESULTS AND DISCUSSION

### 4.1 DEPENDENCY GAP

The best model for each task for each architecture was evaluated on a test set containing examples with a variety of dependency gaps. Figure 1 shows the effect of dependency gap on accuracy on these test examples, with each task represented as a differently coloured line. A dotted vertical line indicates the maximum dependency gap seen in training examples. We can see that, with the exception of Mamba on the period task and the Transformer on the rev task, larger dependency gaps are not associated with a notable decrease in performance, and performance does not differ substantially between test examples whose lengths are within the training distribution and those whose lengths are out-of-distribution. This is somewhat unexpected as this is implicitly a major focus of much of the work studying long-range dependencies. In particular, in the case of modelling natural language, there are many scenarios in which we are concerned with the dependency gap over other elements of a dependency – for example, when looking at verbal agreement across a large span of words.

### 4.2 SEQUENCE LENGTH

The performance on the test set was then analysed with respect to sequence length. Figure 2 shows these results, where again the maximum sequence length seen in training is marked with a dotted line. In this case, the relationship between sequence length and accuracy is much more complex and varies across architectures and tasks. We can see that for the period task, the relationship between sequence length and accuracy is not straightforwardly linear, with higher accuracy on some long sequences when compared with shorter ones.[1] We theorise that since, in this case, the sequences are made up of a smaller sequence repeated multiple times this may correspond to the model being familiar with examples where the *repeated* sequence is the same length, even if the overall sequence is longer due to more repetitions. Most models perform well on the sort task, generalising well to out-of-distribution sequence lengths. The even and odd tasks proved difficult, with all models failing to generalise beyond the distribution for the even task and several failing to learn the odd task even within the distribution. Otherwise, the general pattern across tasks is a decline in performance for

---

[1]We note, however, that the graph appears to indicate 100% accuracy for some longer target sequence lengths – in many cases this is in fact because the target sequence length is a prime number, and thus is not represented in the test dataset, since the target sequences consist of repeated sequences.

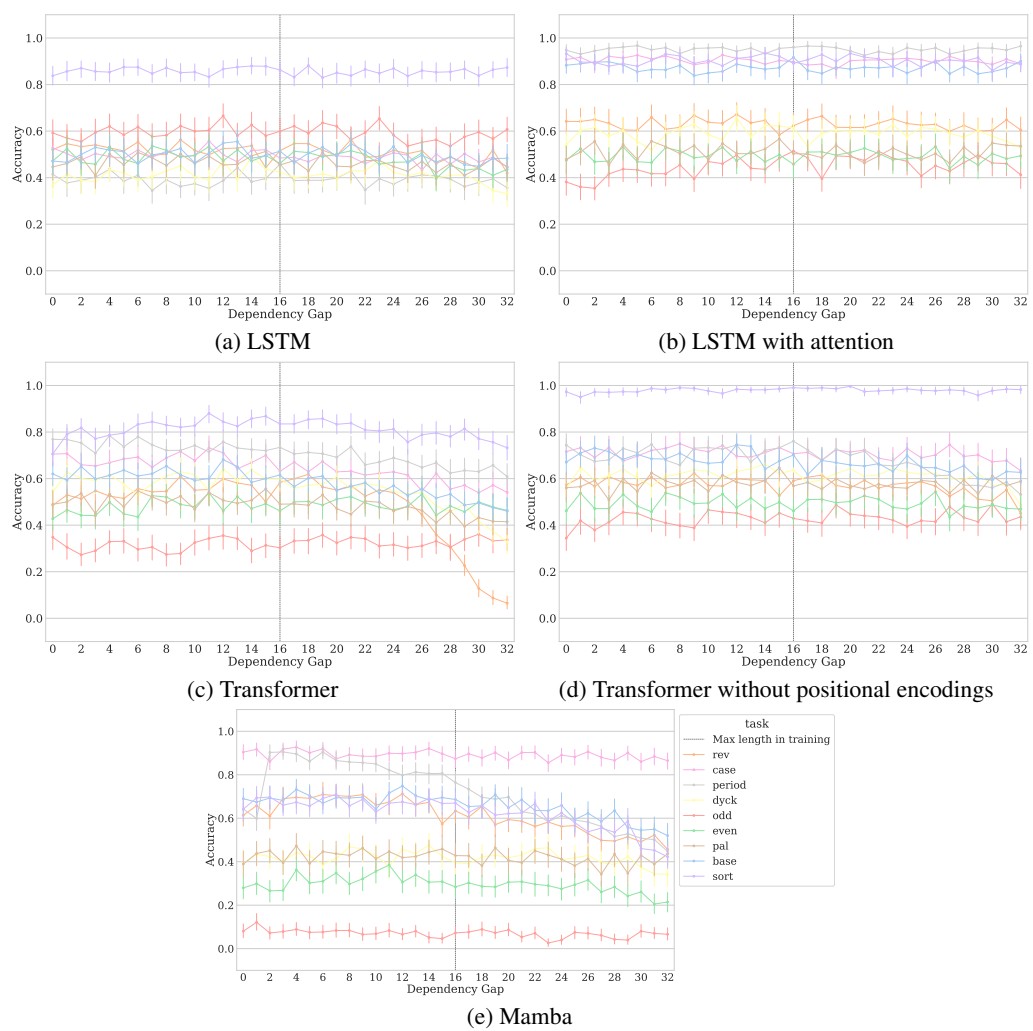

Figure 1: Effect of dependency gap on accuracy

lengths longer than those seen in training. Further variation between architectures and tasks will be discussed in section 4.3.

We questioned whether this pattern could be due to increased memory requirements for longer sequences. We theorise that if that is the case, we would expect to see improvements in performance on long sequences for larger models with more parameters. In appendix B.1 we show the relationship between the number of model parameters and model performance across sequence lengths. The dashed white line indicates the maximum sequence length seen in training. We can see that there is generally not a clear relationship between the number of parameters and the model's ability to handle longer sequences. In some tasks there appears to be a threshold below which the model fails to learn the task at all, but we fail to see a clear linear relationship whereby an increase in available parameters straightforwardly extends the length of the sequences that the model can handle accurately. Thus we conclude that the decrease in performance is the result of difficulty due to processing more sequence elements rather than a memory limitation.

## 4.3 PERFORMANCE ACROSS TASKS

We can consider each task individually and compare the performance of each architecture. Results for each task are shown in table 1, broken down into accuracy on in-distribution and out-of-distribution examples. Here, out-of-distribution examples are those whose dependency gap *or* target sequence length are longer than the examples in the training set. Additionally appendix B.2 shows model

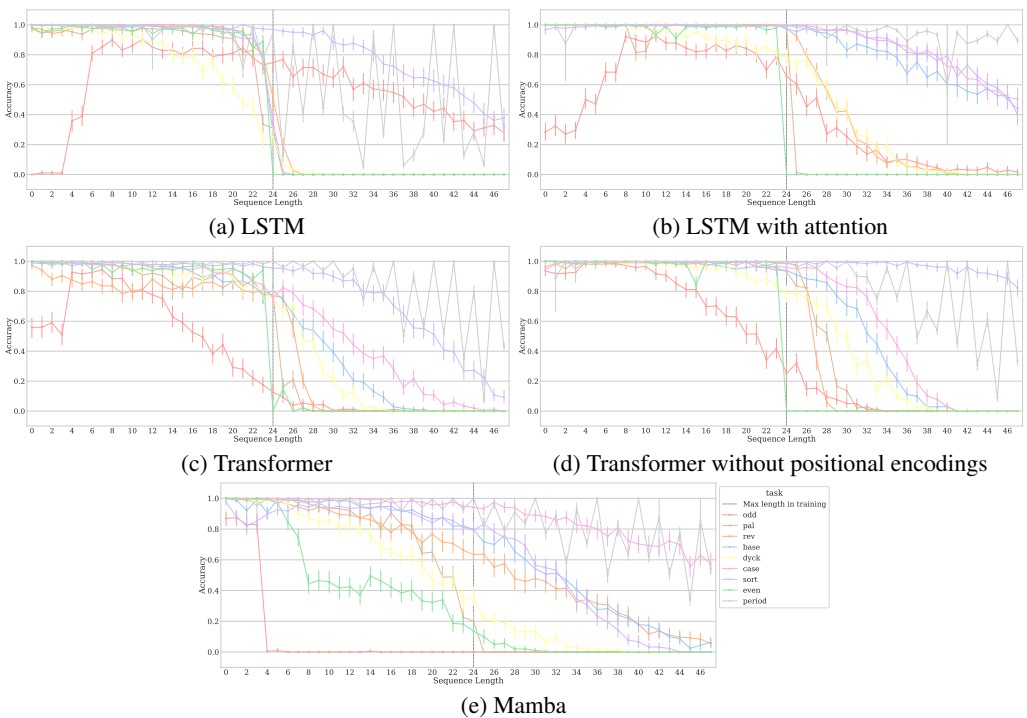

Figure 2: Effect of sequence length on accuracy

performance on each task broken down only by target sequence length, which has proven to be the limiting factor for all models.

On the base task all models succeed in achieving high accuracy for in-distribution examples, however performance declines for tasks with out-of-distribution target sequence length. The decline is the least pronounced for the LSTM with attention, which maintains some ability to perform the task for target sequences of all lengths.

For the rev task, not all models achieved high accuracy even on in-distribution examples. The Mamba model sees declining performance as target sequence length increases even within the training distribution, but this decline does not accelerate for lengths that are out-of-distribution, which results in the Mamba model achieving the best performance on the longest of the out-of-distribution sequences.

The even task involves performing either the base or the rev task depending on whether the length of the target sequence is even or odd, so we might expect to see similar performance on this task and on the previous two tasks. However, on this task all models fail to accurately perform the task on out-of-distribution target sequence lengths. Given that we know that these models are capable of performing the base and rev tasks on out-of-distribution sequences, it seems likely that models simply memorise which task corresponds with which target sequence length without distilling the generalisable insight that this depends on whether or not this length is even, and thus fail when presented with sequence lengths that have not been seen in training.

On the odd task, no model performs perfectly even on in-distribution examples. The Mamba model seems to fail to learn this task entirely. The Transformer without positional encodings achieves the best in-distribution performance, but this declines substantially on out-of-distribution examples. The LSTM struggles with some of the shortest in-distribution sequences, but does not exhibit as much of a decline for longer sequences and thus achieves performance on out-of-distribution sequences that nearly matches its performance on in-distribution examples.

For the sort task, all models achieve good performance on in-distribution sequence lengths, and, with the exception of the Mamba models, the out-of-distribution performance is also generally good. The Transformer without positional encodings exhibits particularly impressive performance on long sequences here. This aligns with what was observed by Delétang et al. (2023) on sorting tasks.

|  |  | LSTM | LSTM with attention | Transformer | Transformer without positional encodings | Mamba |
|---|---|---|---|---|---|---|
| base | In-distribution | 98.29 | 99.87 | 99.33 | 99.33 | 96.23 |
|  | Out-of-distribution | 35.75 | 83.53 | 46.35 | 57.73 | 57.02 |
| rev | In-distribution | 98.70 | 100.0 | 99.53 | 100.00 | 93.81 |
|  | Out-of-distribution | 37.92 | 52.78 | 34.03 | 45.22 | 51.94 |
| even | In-distribution | 99.72 | 99.06 | 99.48 | 99.72 | 61.79 |
|  | Out-of-distribution | 34.24 | 35.95 | 34.53 | 35.81 | 20.91 |
| odd | In-distribution | 66.45 | 70.68 | 62.17 | 80.04 | 15.31 |
|  | Out-of-distribution | 57.28 | 37.64 | 23.87 | 32.70 | 4.74 |
| sort | In-distribution | 99.41 | 99.86 | 99.91 | 99.86 | 95.58 |
|  | Out-of-distribution | 81.90 | 87.60 | 75.82 | 97.39 | 53.05 |
| pal | In-distribution | 94.42 | 99.91 | 98.99 | 99.31 | 87.33 |
|  | Out-of-distribution | 32.95 | 37.46 | 34.40 | 46.12 | 29.49 |
| case | In-distribution | 97.60 | 99.91 | 98.48 | 99.08 | 99.08 |
|  | Out-of-distribution | 35.72 | 87.66 | 54.96 | 62.86 | 86.42 |
| period | In-distribution | 91.05 | 99.51 | 98.90 | 99.51 | 98.28 |
|  | Out-of-distribution | 34.55 | 94.64 | 68.23 | 65.11 | 68.85 |
| dyck | In-distribution | 82.64 | 94.60 | 97.88 | 96.31 | 79.09 |
|  | Out-of-distribution | 29.67 | 49.07 | 43.71 | 51.3 | 31.41 |

Table 1: In- and out-of-distribution accuracy for best models of each architecture

The case task is, in effect, the same as the base task but with the addition of a mapping between upper and lower case mappings. Accordingly, the performance on this task largely mirrors the trends seen for the base task.

The period task is largely performed well in-distribution by all models. The LSTM with attention performs well on this task across all target sequence lengths, including those that are out-of-distribution. Other models display a pattern on this task that is distinct from other tasks, with performance by target sequence length not exhibiting a clear relationship. This may suggest that the critical element for this task was not the target sequence length, but the length of the repeated sequence within this target sequence. The variable performance may correspond to whether or not the model had been trained on *repeated sequences* of the same length, and thus there is no discernible relationship with the overall target sequence length.

The similar pal and dyck tasks prove difficult for all models, with the LSTM and Mamba models struggling even across target sequences whose lengths are in-distribution.

The uneven performance across models and tasks demonstrates the particular difficulty in categorising these tasks in a meaningful way. Although some tasks are performed poorly by all models or well by all models, there does not appear to be a uniform ranking of task difficulty that holds true across architectures, as might be suggested by characterising task complexity in terms of the Chomsky hierarchy, for example. Instead it is possible that architectures with particular inductive biases may find some tasks easier to learn while architectures with other inductive biases might find the same task more difficult. Further investigation is certainly warranted into how these tasks might be characterised in a way that is useful and informative in terms of predicting model performance.

## 4.4 ATTENTION AND POSITION ENCODING

We sought to investigate how the addition of an attention mechanism might affect the performance of an LSTM-based architecture. In fig. 3a we highlight the tasks where a substantial difference in performance was observed. Although the addition of an attention mechanism seems to improve performance on the base, case and period tasks, allowing for better performance on longer sequences. These tasks are all fundamentally copying-based tasks, so this performance improvement is likely due to the attention mechanism enabling better copying of long sequences. On the other hand, the attention mechanism seems to degrade performance somewhat on longer sequences for the odd task. This task can be approached by removing any character the second time it occurs, so it can be approached by counting without needing to look back and copy long sequences, which may explain why attention is not beneficial.

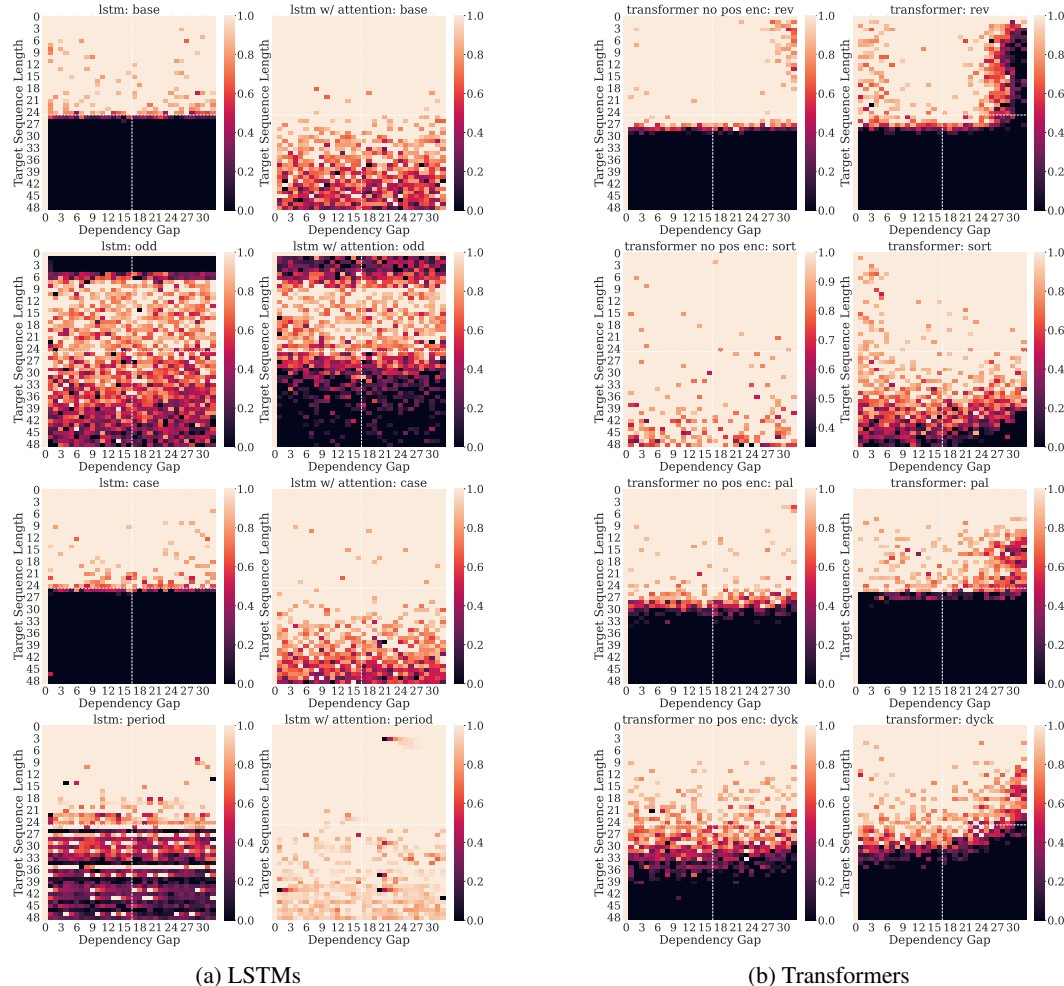

(a) LSTMs          (b) Transformers

Figure 3: Performance differences due to inclusion of attention in LSTM and position encodings in transformers

Similarly, we made a comparison of performance between transformers with and without position encoding. Differences here were generally small, and the tasks for which there was some difference are shown in fig. 3b. The removal of position encodings seems to confer a small advantage when dealing with longer dependency gaps in the rev, pal and dyck tasks. It is unclear what element of these tasks may explain this. Their removal also seems to improve performance on longer sequence lengths in the sort task. The positions of the target sequence tokens is irrelevant here, so it is not surprising that the position encodings do not improve performance.

## 5 RELATED WORK

A prominent benchmark for long-range dependencies was presented by Tay et al. (2021). They created a series of tasks focused on evaluating models' abilities to model these dependencies. These tasks focus predominantly on tasks whose solutions require integrating knowledge from across a long span of context. They don't explicitly categorise how the tasks may differ in terms of the complexity of the processing required, the number of dependencies being considered at once, or the distance within a sequence between a token and tokens upon which it depends. In previous work, Delétang et al. (2023) carried out a systematic evaluation of architectures on tasks focused on how these tasks correspond to levels of the Chomsky hierarchy. In this work, although our tasks contain similar elements, the nature of the tasks is only one element rather than the sole focus, with the additional elements of variable sequence length and dependency gap.

An alternative way of characterising task difficulty is used by Dziri et al. (2023), who assess the ability of transformers to carry out compositional tasks. They do this by formulating tasks in terms of computation graphs, whose properties can then be used to characterise the difficulty of the task. For example, they consider the maximal width of the computation graph, and the length of paths through the graph. This presents an alternative way to think about the difficulty of tasks which may offer more nuance than considering them in terms of formal languages, allowing for a more unified consideration of processing complexity and memory requirements.

Supporting some of our findings, Jelassi et al. (2024) carry out experiments that demonstrate that transformers perform better than Mamba models at tasks involving copying from input context. They also justify theoretically why this may be the case, and suggest that endowing Mamba models with attention mechanisms or external longer-term memory may remedy this deficiency.

## 6    CONCLUSION

In this work, we focused on long-range dependencies in language data. We sought to define what constitutes a long-range dependency and to clearly characterise dependencies based on their properties. Aligning with this characterisation, we devised a framework for systematic examination of model performance on long-range dependencies, varying the number of tokens involved in the dependency, the gap between a token and the tokens on which it depends and the nature of the dependency between them. Using this framework, we evaluated and analysed a variety of architectures. Our findings indicate that the gap between a token and the tokens on which it depends is potentially substantially less important than other elements such as the number of tokens involved and how complex the dependency is. This may indicate the need for a refocusing of research into models' performance on dependency-based tasks. We also find that it is not uniformly the case that some architectures or architectural elements are superior for the handling of long-range dependencies, and that in fact this may depend more on the nature of the dependency in question. This is a key element to consider when discussing the inductive biases of architectures. Characterising the functions that define a dependency in terms of complexity is difficult, with a number of different frameworks being used in research. Additionally, in real-world settings, dependencies are rarely as straightforward as the tasks examined here. This motivates future research to comprehensively characterise the complexity of these tasks and to develop an understanding of how dependencies manifest in real-world data.

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
