# OpenReview forum: "It Depends: Understanding Why Models Struggle with Long-Range Dependencies"
_ICLR.cc/2026/Conference — Submitted to ICLR 2026_

### Official Review · Reviewer_aiyT · 2025-10-30

**Soundness:** 1
**Presentation:** 1
**Contribution:** 2
**Rating:** 2
**Confidence:** 4

**Summary:**

This paper studies the factors that affect the long-range dependency performance of language models. They start by properly defining the term "long-range dependency" and introduce several types of long-range dependency with different levels of complexity. They train language models (LMs) of different architectures and evaluate those LMs on in-distribution and out-of-distribution dependency datasets. They find that the distance of the dependency is a less critical factor, compared with the length of the target sequence (number of tokens involved in the dependency) and the complexity of the dependency.

**Strengths:**

- This paper introduces interesting experiment settings to study LM's ability to model long-range dependency.
- The experiment setting and definition of the problem of focus are clearly defined.
- The architectures of the LM studied in this paper cover many important sequence modeling models.
- The conclusion is somewhat interesting, if neglecting some questionable experiment settings

**Weaknesses:**

- The experiment setting is odd. All models, even the same model architecture, have a different model size. It is unclear why the same model architecture has a different model size for different types of dependency. This makes me question whether the different performance of different dependencies is due to the model architecture.
- Limited experiment evaluation. The length explored in this paper, even in the OOD case, is only 32. This is very different from the context window of current LMs, which can be up to a few thousand. The dependency length studied in this paper is too small and may be hard to generalize to longer sequences.
- Insufficient related works and citations. The paper only contains 6 references. This is insufficient and unacceptable. Many papers that should be cited, including LSTM and transformers, which are the models used in this paper. Moreover, using synthetic language with different difficulties to fine-tune LMs has already been widely explored, and can be dated back to at least 2020. However, none of those papers are discussed.
- Lack of explanations on why a specific model architecture works well or poorly on a specific type of dependency. It feels like this paper only provides the observation but lack more detailed rationales or insights.
- Poor visualization. The legends (colors) in Figures 1 and 2 are almost unreadible when printing the manuscript on A4 paper. It is hard to distinguish which line corresponds to which dependency.

**Questions:**

I found many ungrammatical sentences and missing commas in the paper. The authors are encouraged to proofread the paper.
Examples:
- Line 15: In this work '**,**'
- Line 18: these factors '**,**' we
- Multiple places: i.e. "**,**"
- Line 372: "*Although the addition of an attention mechanism seems to improve performance on the base, case and period tasks, allowing for better performance on longer sequences*" $\to$ Missing a main verb

---

### Official Review · Reviewer_toJa · 2025-11-01

**Soundness:** 2
**Presentation:** 2
**Contribution:** 2
**Rating:** 2
**Confidence:** 4

**Summary:**

This paper proposes a framework for analyzing long-range dependencies and examining how various factors influence architectural performance. Experiments reveal that the distance between a token and its dependent tokens has only a limited effect on model performance, while factors such as the number of tokens involved, the complexity, and the type of dependency play a more significant role. The results also show that architectural components do not have uniform effects across tasks; their impact varies depending on the nature of the dependency being modeled.

**Strengths:**

1. This paper introduces a synthetic framework for evaluating models on tasks involving long-range dependencies. The framework enables detailed cross-model comparisons and facilitates precise analyses of how various factors, such as dependency distance, impact performance.
2. The paper yields several interesting findings: The distance between a token and its dependent tokens has only a limited effect on model performance, while factors such as the number of tokens involved, the complexity, and the type of dependency play a more significant role. The results also show that architectural components do not have uniform effects across tasks; their impact varies depending on the nature of the dependency being modeled.

**Weaknesses:**

1. In the introduction (presented without a section title), the authors highlight the ambiguity in how prior research defines "long-range dependencies." They raise questions such as whether the difficulty arises purely from the distance between dependent elements, or whether it also depends on the complexity of the dependency, and whether different architectures exhibit distinct challenges with these dependencies. However, the paper lacks a substantive review of existing literature on long-range dependencies, including prior definitions and explanations of why this question remains unresolved in current research.
2. Some of the experimental claims appear to be overstated. For instance, the authors assert in the introduction that "*the effect on performance was not alleviated by increasing model size,*" yet no experiments are presented that explicitly analyze the influence of model size on performance.
3. The scope of the proposed method is restricted to the character level, which limits its applicability to real-world scenarios that primarily operate at the word level. In addition, the architectures evaluated, such as LSTMs, are not representative of contemporary language models in practical use.

**Questions:**

See "Weaknesses."

---

### Official Review · Reviewer_ftLh · 2025-11-01

**Soundness:** 2
**Presentation:** 2
**Contribution:** 2
**Rating:** 2
**Confidence:** 3

**Summary:**

The authors seek to understand the long-context modeling capabilities of different network architectures through training experiments on synthetic language tasks. Authors conclude that task type and sequence length are the two most important factors impacting long-context modeling capabilities with different architectures struggling at different sequence lengths for different tasks. The content of the paper is minimal and the authors are missing an appendix with more details on the experiments and results.

**Strengths:**

The paper provides a minimal study on long context modeling capabilities of common language modeling neural network architectures on synthetic language tasks. The framework for generating synthetic long-context tasks is an interesting approach to stress testing model capabilities.

**Weaknesses:**

Unfortunately, it appears that the appendix is missing from the paper and I am unable to evaluate the model scaling experiments and other analysis performed in the paper. Without the appendix, the content of the paper is a bit sparse. My general impression is that while the findings are interesting, the paper is a bit light on content and doesn't present a novel contribution to long-context modeling research. With the current experiment setup, I don't think the authors have enough data points to draw conclusions about the relative capabilities of different architectures and would require scaling laws to estimate the performance of different architectures at larger scales.

**Questions:**

1. Since the training data is pretty small scale it would be useful to investigate how scaling data tokens and model parameters in conjunction affects long-context modeling capabilities.
2. How large are the models used in the experiments? Are you using the same number of parameters for all architectures?
3. Have you experimented with how pre-trained language models would perform on the synthetic long-context tasks? As language models are pre-trained on large-scale data prior to task adaptation, would this initialization change the relative capabilities of different architectures?

---

### Official Review · Reviewer_9FXV · 2025-11-01

**Soundness:** 2
**Presentation:** 2
**Contribution:** 2
**Rating:** 2
**Confidence:** 4

**Summary:**

The paper aims to challenge the inconsistent use of the term "long-range dependency" in the literature and to propose a synthetic framework for analyzing and disentangling the factors that impact model performance. The factors considered in the paper are architecture, dependency gap, target sequence length, and complexity of the dependency (function).

The authors do this by creating a synthetic framework in an artificial language, allowing the factors considered in the paper to be explicitly controlled. Experiments are then conducted to study the effect of the Cartesian set of the proposed factors.

One of the main findings of the paper is that dependency gaps do not influence performance substantially (across all architectures). The authors further find that the complexity of the dependency function and length of the dependency itself (or target length) matter far more; performance decreases sharply with increasing complexity and dependency length; and these findings are not affected by model size.

The experiments also show that model architecture, attention, and positional embeddings substantially affect task performance.

**Strengths:**

The paper provides a clear, quantifiable definition of long-range dependency and a synthetic framework that allows explicit manipulation of disentangled factors: the dependency gap, dependency/target length, and dependency complexity (ranging from simple copying to complex even- and odd-numbered functions defined in the paper).

The architecture families (LSTMs with and without attention, Transformers with and without positional embeddings, and Mamba) evaluated on the tasks are comprehensive. The evaluation of OOD sequence lengths is also a good test of generalization.

The authors also present experiments demonstrating that their findings on dependency length and complexity are not affected by model size.

**Weaknesses:**

The paper's main claim—that dependency gaps do not affect model performance — is, in my understanding, unsubstantiated. The experiments only evaluate dependency gaps of length up to 48; this is extremely small compared to the usual long-range dependency benchmarks existing in the literature: Tay et al., 2021, studying between 1K and 16K tokens, and other future language long-range/long-context benchmarks [1,2,3] studying 1k+ tokens as well (up to 200k). Given the limited scope of evaluation presented, it is not clear if these results would translate to longer dependency/context cases.

Furthermore, the experiments have been performed on a synthetically crafted dataset and benchmark, without providing evidence of why these findings would translate to natural language data. An experiment on natural language data, with samples divided along similar factors as in the synthetic dataset, is necessary to provide evidence for the soundness of the synthetic setup.

As an extension of the previous comment, most of the functions used for dependencies do not have parallels in natural language (even, odd, sort), and the tasks do not obey the Chomsky hierarchy as well as pointed out in the paper (lines 361-362). Therefore, these findings cannot be studied by future works or evaluated by future models on real data.

The paper also makes a very strong assumption of there being a single dependency per sample and a single task per model, which is not the case in natural language modeling. The findings that the dependency gap does not affect performance in the experiments (Fig. 1), may be an artifact of this assumption.

The actual models used from the specified model families are also extremely small (up to a maximum of 4 transformer or LSTM layers). The length of the training is not specified in the paper. Were all models trained to convergence? Was this epoch/number of training samples value common to all setups, or unique to each setup? Why does the size of the model (provided in the Appendix) vary so highly across tasks and architectures?

The paper aims to challenge the existing vague notion of long-range dependency in the literature. However, the related work section is presented only at the end of the paper and lacks depth. In this specific case, challenging the status quo, it is, in my opinion, important to present the status quo and explain how the proposed methodology differs for a proper perspective. No comprehensive review of long-range and long-context benchmarks from previous literature [1,2,3] has been provided. The paper does not present any related work in the area of benchmarking language or non-language models, nor does it contrast or support its methodology or findings. I am not an expert in language model benchmarking, but the paper has only six citations, given its position in active areas of research (model benchmarking, long-range dependency evaluation, language model evaluation), suggesting a lack of comprehensiveness in its background and related work presentation. Please see papers cited below for a non-exhaustive list of relevant papers.

Presentation issues:
1. The plots are very hard to read, given the thin lines, very close colors, and only one legend on the figure, which is far from the plot, making comparisons hard.

Non-exhaustive list of missing relevant references:

[1] Ni, X., Cai, H., Wei, X., Wang, S., Yin, D., & Li, P. (2024). XL $^ 2$ Bench: A Benchmark for Extremely Long Context Understanding with Long-range Dependencies. arXiv preprint arXiv:2404.05446.

[2] Liu, X., Dong, P., Hu, X., & Chu, X. (2024). Longgenbench: Long-context generation benchmark. arXiv preprint arXiv:2410.04199.

[3] Yuan, T., Ning, X., Zhou, D., Yang, Z., Li, S., Zhuang, M., ... & Wang, Y. (2024). Lv-eval: A balanced long-context benchmark with 5 length levels up to 256k. arXiv preprint arXiv:2402.05136.

[4] Wang, C., Duan, H., Zhang, S., Lin, D., & Chen, K. (2024). Ada-leval: Evaluating long-context llms with length-adaptable benchmarks. arXiv preprint arXiv:2404.06480.

[5] Hsieh, C. P., Sun, S., Kriman, S., Acharya, S., Rekesh, D., Jia, F., ... & Ginsburg, B. (2024). RULER: What's the Real Context Size of Your Long-Context Language Models?. arXiv preprint arXiv:2404.06654.

[6] Bai, Y., Lv, X., Zhang, J., Lyu, H., Tang, J., Huang, Z., ... & Li, J. (2023). Longbench: A bilingual, multitask benchmark for long context understanding. arXiv preprint arXiv:2308.14508.

[7] Liu, N. F., Lin, K., Hewitt, J., Paranjape, A., Bevilacqua, M., Petroni, F., & Liang, P. (2023). Lost in the middle: How language models use long contexts. arXiv preprint arXiv:2307.03172.

**Questions:**

Do the findings of the paper scale to larger dependency gaps (>100, >1000)? Do these findings still stand in the case of a single model handling multiple tasks (standard in modern self-supervised language models)?

Why/how would these findings on the proposed synthetic dataset translate to a real-world natural language dataset?

How does the paper set itself apart from existing work in the field done on real datasets? What value does it add?

---

### Meta-Review · Area_Chair_aHze · 2026-01-05

**Summary:**

The experiments are largely incomplete. The main claim is that dependency gaps do not affect model performance, but the maximum gap length they test is much smaller than usually considered an issue. The experiments they do have are synthetic and the reviewers expressed points of confusion around what models were used and how size was calculated. The results that are provided are not analyzed in detail.

**Reviewer Concerns:**

No rebuttal.

**Reviewer Scores:**

Scores would be unchanged due to lack of rebuttal.

---

### Decision · Program_Chairs · 2026-01-26

Reject